# Accurate Point Cloud Registration
# with Robust Optimal Transport

**Zhengyang Shen**[*]
UNC Chapel Hill
zyshen@cs.unc.edu

**Jean Feydy**[*]
Imperial College London
jfeydy@ic.ac.uk

**Peirong Liu**
UNC Chapel Hill
peirong@cs.unc.edu

**Ariel Hernán Curiale**
Harvard Medical School
acuriale@bwh.harvard.edu

**Ruben San José Estépar**
Harvard Medical School
rubensanjose@bwh.harvard.edu

**Raúl San José Estépar**
Harvard Medical School
rjosest@bwh.harvard.edu

**Marc Niethammer**
UNC Chapel Hill
mn@cs.unc.edu

## Abstract

This work investigates the use of robust optimal transport (OT) for shape matching. Specifically, we show that recent OT solvers improve both optimization-based and deep learning methods for point cloud registration, boosting accuracy at an affordable computational cost. This manuscript starts with a practical overview of modern OT theory. We then provide solutions to the main difficulties in using this framework for shape matching. Finally, we showcase the performance of transport-enhanced registration models on a wide range of challenging tasks: rigid registration for partial shapes; scene flow estimation on the Kitti dataset; and nonparametric registration of lung vascular trees between inspiration and expiration. Our OT-based methods achieve state-of-the-art results on Kitti and for the challenging lung registration task, both in terms of accuracy and scalability.

We also release PVT1010, a new public dataset of 1,010 pairs of lung vascular trees with densely sampled points. This dataset provides a challenging use case for point cloud registration algorithms with highly complex shapes and deformations. Our work demonstrates that robust OT enables fast pre-alignment and fine-tuning for a wide range of registration models, thereby providing a new key method for the computer vision toolbox. Our code and dataset are available online at: https://github.com/uncbiag/robot.

## 1 Introduction

Shape registration is a fundamental but difficult problem in computer vision. The task is to determine plausible spatial correspondences between pairs of shapes, with use cases that range from pose estimation for noisy point clouds [14] to the nonparametric registration of high-resolution medical images [17]. As illustrated in Fig. 1, most existing approaches to this problem consist of a combination of three steps, possibly fused together by some deep learning (DL) methods: (1) feature extraction; (2) feature matching; and (3) regularization using a class of acceptable transformations that is specified through a parametric or nonparametric model. This work discusses how tools derived from

---

[*]Equal Contribution.

35th Conference on Neural Information Processing Systems (NeurIPS 2021).

optimal transport (OT) theory [87] can improve the second step of this pipeline (feature matching) on challenging problems. To put these results in context, we first present an overview of related methods.

**1. Feature extraction.** To establish spatial correspondences, one first computes descriptive local features. When dealing with (possibly annotated) point clouds, a simple choice is to rely on Cartesian coordinates $(x, y, z)$ [3, 26]. Going further, stronger descriptors capture local geometric and topological properties: examples include shape orientation and curvatures [21, 96], shape contexts [6], spectral eigenvalues [84, 70] and annotations such as color [76] or chemical fingerprints [43, 113]. Recently, expressive feature representations have also been *learned* using deep neural networks (DNN): see [16] and subsequent works on *geometric* deep learning. Generally, feature extractors are designed to make shape registration as unambiguous as possible. In order to get closer to the ideal case of landmark matching [11], we associate discriminative features to the salient points of our shapes: this increases the robustness of the subsequent *matching* and *regularization* steps.

**2. Feature matching.** Once computed on both of the source and target shapes, feature vectors are put in correspondence with each other. This assignment is often encoded as an explicit mapping between the two shapes; alternatively, the vector field relating the shapes can be defined implicitly as the gradient of a geometric loss function that quantifies discrepancies between two distributions of features [35]:

a) A first major approach is to rely on **nearest neighbor** projections and the related chamfer [12] and Hausdorff distances [13], as in the Iterative Closest Point (ICP) algorithm [7]. This method can be softened through the use of a softmax (log-sum-exp) operator as in the many variants of the Coherent Point Drift (CPD) method [81, 73, 72, 44], or made robust to outliers in the specific context of rigid and affine registrations [41, 129, 128, 14].

b) Alternatively, a second approach is to rely on **convolutional kernel norms** such as the Energy Distance [94], which are also known as Maximum Mean Discrepancies (MMD) in statistics [48]. These loss functions are common in imaging science [88] and computational anatomy [115, 21] but are prone to vanishing gradients [40, 39].

c) Finally, a third type of approach is to rely on **optimal transport (OT)** theory [87] and solutions of the earth mover's problem [97]. This method is equivalent to a nearest neighbor projection under a global constraint of bijectivity that enforces consistency in the matching. On the one hand, OT has been known to provide reliable correspondences in computer vision for more than two decades [26, 47, 64]. On the other hand, it has often faced major issues of scalability and robustness to outliers on noisy data. As detailed below, the main purpose of this work is to overcome these limitations and enable the widespread use of OT tools for challenging registration problems.

**3. Regularization with a deformation model.** The output of the two steps above is a non-smooth vector field that may not be suitable for downstream tasks due to e.g. tears and compression artifacts. As a third step, most registration methods thus rely on regularization to obtain plausible deformations. This process is task-specific, with applications that range from rigid registration [133, 46, 30, 2, 121, 122] to free-form motion estimation [91, 126, 69, 49]. In Sec. 3, we address the interaction of OT matching layers with a varied collection of regularization strategies – from optimization-based spline and diffeomorphic models to DNNs.

**Recent progresses.** Research works on shape registration combine ideas from the three paragraphs above to best fit the characteristics of computer vision problems [71, 31, 107]. Over the past few years, significant progress has been made on all fronts. On the one hand, (geometric) deep learning networks have been used to define data-driven feature maps [92, 92, 123] and multiscale regularization modules [126, 68, 108], sometimes fused within end-to-end architectures [133, 91, 132, 30]. On the other hand, nearest neighbor projections, kernel convolutions and transport-based matching strategies have all been generalized to take advantage of these modern descriptors: they can now be used in high-dimensional feature spaces [59, 37].

**Challenges.** Nevertheless, state-of-the-art (SOTA) methods in the field still have important limitations. First, modern deep learning pipelines are often hard to train to "pixel-perfect" accuracy on non-smooth shapes, with diminishing returns in terms of model size and training data [2]. Second, scaling up point neural networks to finely sampled shapes (N > 10k points) remains a challenging research topic [49, 135, 37]. Third, the impact of the choice of a specific feature matching method on the performance of deep learning models remains only partially understood [58].

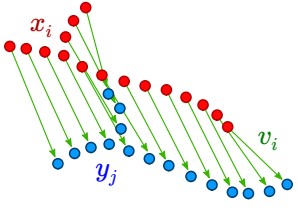 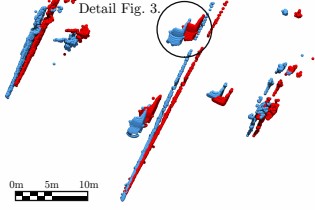 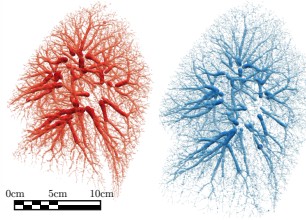

(a) Source and target points in dimension D = 2.

(b) Source and target frames, sampled with 30k points each.

(c) Exhale and inhale trees, sampled with 60k points each.

Figure 1: **Robust Optimal Transport (RobOT)** generalizes sorting to spaces of dimension $D \geqslant 1$. (a) RobOT is equivalent to a nearest neighbor projection subject to mass distribution constraints that make it robust to translations and small deformation. We demonstrate that RobOT is now ready to be part of the standard toolbox in computer vision with extensive numerical experiments for 3D scene flow estimation (b) and lung registration (c). Rendering done with Paraview [1] and PyVista [112].

**Related works.** Following major progress on computational OT in the mathematical literature [79, 27, 66, 103], improved modules for feature matching have attracted interest as a possible solution to these challenges. Works on sliced partial OT [9] and dustbin-OT [28] have shown that outliers can be handled effectively by OT methods for rigid registration, beyond the classic Robust Point Matching method (RPM) [47, 26]. Going further, the Sinkhorn algorithm for entropy-regularized OT [27, 64, 104, 65] has been studied extensively for shape registration in computational anatomy [36, 45] and computer graphics [74, 33, 85]. The Gromov–Wasserstein distance has also been used for shape analysis [110, 118], albeit at a higher computational cost. These applications have driven interest in the development of a complete theory for **Robust Optimal Transport** (RobOT), outlined in Sec. 2, which handles both sampling artifacts and outliers [25, 105, 67, 24, 106, 80]. Most recently, this framework has started to be used in shape analysis with applications to shape matching [36], the segmentation of brain tractograms [38] and deep deformation estimation with the FLOT architecture [91].

**Contributions.** We build upon the work above to tackle challenging point cloud registration problems for scene flow estimation and computational anatomy. Our **key contributions** are:

1. **Accurate feature matching with scalable OT solvers.** For the first time, we scale up RobOT for deep feature matching to high-resolution shapes with more than 10k points. To this end, we leverage the latest generation of OT solvers [39, 35] and overcome significant issues of memory usage and numerical stability. This allows us to handle fine-grained details effectively, which is key for e.g. most medical applications.

2. **Interaction with task-specific regularization strategies.** We show how to interface RobOT matchings with advanced deformation models. This is in contrast with e.g. the FLOT architecture, which focuses on the direct prediction of a vector field and cannot be used for applications that require guarantees on the smoothness of the registration.

3. **Challenging new dataset.** We release a large dataset of lung vascular trees that should be registered between inhalation and exhalation. This relevant medical problem involves large and complex deformations of high-resolution 3D point clouds. As a new benchmark for the community, we provide two strong baselines that rely respectively on global feature matching and on deep deformation estimation.

4. **Consistent SOTA performance.** Our proposed models achieve SOTA results for scene flow on Kitti [78, 77] and for point-cloud-based lung registration on DirLab-COPDGene [19]. Notably, we show that RobOT is highly suited to fine-tuning tasks: it consistently turns "good" matchings into nearly perfect registrations at an affordable numerical cost.

**Main experimental observations.** Is OT relevant in the deep learning era? To answer this question decisively, we perform extensive numerical experiments and ablation studies. We fully document "underwhelming" results in the Supplementary Material and distill the key lessons that we learned in the Deep-RobOT architecture (Section 3.2). This model relies on fast RobOT layers to cover for the main weaknesses of point neural networks for shape registration. It is remarkably easy to deploy and generalizes well from synthetic training data to real test samples. We thus believe that it will have a stimulating impact on both of the computer vision and medical imaging literature.

## 2 Robust optimal transport

This section introduces the mathematical foundations of our work. After a brief overview of Robust Optimal Transport (RobOT) theory, we discuss the main challenges that one encounters when using this framework for computer vision. To avoid memory overflows and numerical stability issues, we introduce the weighted "RobOT matching": a vector field that summarizes the information of a full transport plan with a linear memory footprint. As detailed in the next sections, this representation lets us scale up to high-resolution shapes without compromising on accuracy.

### 2.1 Mathematical background

**The assignment problem.** If $A = (x_1, \ldots, x_N)$ and $B = (y_1, \ldots, y_M)$ are two **point clouds** in $\mathbb{R}^3$ with $N = M$, the assignment problem between A and B reads:

$$\text{Assignment}(A, B) = \min_{s:[\![1,N]\!] \to [\![1,N]\!]} \frac{1}{2N} \sum_{i=1}^{N} \|x_i - y_{s(i)}\|_{\mathbb{R}^3}^2 , \quad \text{where } s \text{ is a permutation.} \quad (1)$$

This problem generalizes sorting to $\mathbb{R}^3$: if the points $x_i$ and $y_j$ all belong to a line, the optimal permutation $s^*$ corresponds to a non-decreasing re-ordering of the point sets A and B [87].

**Robust optimal transport.** Further, OT theory allows us to consider problems where $N \neq M$. **Non-negative weights** $\alpha_1, \ldots, \alpha_N, \beta_1, \ldots, \beta_M \geqslant 0$ are attached to the points $x_i$, $y_j$ and account for variations of the sampling densities, while **feature vectors** $p_1, \ldots, p_N$ and $q_1, \ldots, q_M$ in $\mathbb{R}^D$ may advantageously replace raw point coordinates $x_i$ and $y_j$ in $\mathbb{R}^3$. Following [67, 25], the robust OT problem between the shapes $A = (\alpha_i, x_i, p_i)$ and $B = (\beta_j, y_j, q_j)$ reads:

$$\text{OT}_{\sigma,\tau}(A, B) = \min_{(\pi_{i,j}) \in \mathbb{R}_{\geqslant 0}^{N \times M}} \sum_{i=1}^{N} \sum_{j=1}^{M} \pi_{i,j} \cdot \tfrac{1}{2} \|p_i - q_j\|_{\mathbb{R}^D}^2 \quad (2)$$

$$+ \underbrace{\sigma^2 \, \text{KL}\big(\pi_{i,j} \,\|\, \alpha_i \otimes \beta_j\big)}_{\text{Entropic blur at scale } \sigma.} \; + \; \underbrace{\tau^2 \, \text{KL}\big(\textstyle\sum_j \pi_{i,j} \,\|\, \alpha_i\big)}_{\pi \text{ should match } A \ldots} \; + \; \underbrace{\tau^2 \, \text{KL}\big(\textstyle\sum_i \pi_{i,j} \,\|\, \beta_j\big)}_{\ldots \text{onto } B.} ,$$

for any choice of the regularization parameters $\sigma > 0$ and $\tau > 0$. In the equation above, the Kullback-Leibler divergence $\text{KL}(a_i\|b_i) = \sum a_i \log(a_i/b_i) - a_i + b_i$ is a relative entropy that penalizes deviations of a non-negative vector of weights $(a_i)$ to a reference measure $(b_i)$.

**Parameters.** The first regularization term is scaled by the square of a **blur** radius $\sigma$. This characteristic length quantifies the fuzziness of the probabilistic assignment $(\pi_{i,j})$ between points $x_i$ and $y_j$ [35]. The last two regularization terms promote the matching of the full distribution of points A onto the target shape B: they generalize the constraints of injectivity and surjectivity of Eq. (1) to the probabilistic setting. They are scaled by the square of a maximum **reach** distance $\tau$: this parameter acts as a soft upper bound on the distance between feature vectors $p_i$ and $q_j$ that should be matched with each other [38, 105].

For shape registration, we use simple heuristics for the values of these two characteristic scales: the **blur** $\sigma$ should be equal to the average sampling distance in feature space $\mathbb{R}^D$ while the **reach** $\tau$ should be equal to the largest plausible displacement for any given feature vector $p_i$. These rules are easy to follow if point features correspond to Cartesian coordinates $x_i$ and $y_j$ in $\mathbb{R}^3$ but may lead to unexpected behaviors if features are output by a DNN. In the latter case, we thus normalize our feature vectors so that $\|p_i\|_{\mathbb{R}^D} = \|q_j\|_{\mathbb{R}^D} = 1$ and pick values for $\sigma$ and $\tau$ between 0 and 2.

**Working with a soft, probabilistic transport plan.** As detailed in [35], scalable OT solvers for Eq. (2) return a pair of dual vectors $(f_i) \in \mathbb{R}^N$ and $(g_j) \in \mathbb{R}^M$ that encode **implicitly** an optimal transport plan $(\pi_{i,j}) \in \mathbb{R}^{N \times M}$ with coefficients:

$$\pi_{i,j} = \alpha_i \beta_j \cdot \exp \tfrac{1}{\sigma^2} \big[ f_i + g_j - \tfrac{1}{2} \|p_i - q_j\|_{\mathbb{R}^D}^2 \big] \geqslant 0 . \quad (3)$$

In the limit case where $\sigma$ tends to 0 and $\tau$ tends to $+\infty$, for generic point clouds $(x_i)$ and $(y_j)$ with $N = M$ and equal weights $\alpha_i = \beta_j = 1/N$, $\pi_{i,j}$ is a permutation matrix [87]. We retrieve the simple

assignment problem of Eq. (1): $\pi_{i,j} = 1/N$ if $j = s^*(i)$ and 0 otherwise. However, in general the transport plan must be understood as a probabilistic map between the point distributions A and B that assigns a weight $\pi_{i,j}$ to the coupling "$x_i \leftrightarrow y_j$". For shape registration, this implies that the main difficulties for using robust OT are two-fold: first, the coupling $\pi$ is not one-to-one, but one-to-*many*; second, the lines and columns of the transport plan $\pi$ do not sum up to one. Notably, this implies that when $\tau < +\infty$, the gradient of the OT cost with respect to the point positions $x_i$ is not homogeneous: we observe vanishing and inflated values across the domain [105].

## 2.2 RobOT: a convenient representation of the optimal transport plan

**The weighted RobOT matching.** To work around these issues, we introduce the vector field:

$$v_i = \frac{\sum_{j=1}^{M} \pi_{i,j} \cdot (y_j - x_i)}{\sum_{j=1}^{M} \pi_{i,j}} \in \mathbb{R}^3 \quad \text{with confidence weights} \quad w_i = \sum_{j=1}^{M} \pi_{i,j} \geqslant 0 . \quad (4)$$

This object has the same memory footprint as the input shape A and summarizes the information that is contained in the N-by-M transport plan $(\pi_{i,j})$ – a matrix that is often too large to be stored and manipulated efficiently. This "weighted RobOT matching" is at the heart of our approach and generalizes the standard Monge map from classical OT theory [87] to the setting of (deep) shape registration. In practice, the weighted vector field $(w_1, v_1), \ldots, (w_N, v_N)$ is both convenient to use and easy to compute on GPUs. Let us briefly explain why.

**Fast implementation.** Our differentiable RobOT layer takes as input the two shapes $A = (\alpha_i, x_i, p_i)$ and $B = (\beta_j, y_j, q_j)$, with feature vectors $p_i$ and $q_j$ in $\mathbb{R}^D$ that have been computed upstream using e.g. a point neural network. It returns the N vectors $v_i$ with weights $w_i$ that map the source points $x_i$ onto the targets $y_j$ in $\mathbb{R}^3$. Starting from the input point features $p_i, q_j$ and weights $\alpha_i$, $\beta_j$, we first compute the optimal dual vectors $f_i$ and $g_j$ using the fast solvers of the GeomLoss library [39]. We then combine Eq. (3) with Eq. (4) to compute the RobOT vectors $v_i$ and weights $w_i$ with $O(N + M)$ memory footprint using the KeOps library [20, 37] for PyTorch [86] and NumPy [117]. We use a log-sum-exp formulation to ensure numerical stability. Remarkably, our implementation scales up to $N, M = 100k$ in fractions of a second. Unlike common strategies that are based on dense or sparse representations of the transport plan $\pi$, our approach is perfectly suited to a *symbolic* implementation [37] and streams well on GPUs with optimal, contiguous memory accesses.

**Comparison with nearest neighbor projections.** We use our RobOT layer as a plug-in replacement for closest point matching [7]. The *blur* ($\sigma$) and *reach* ($\tau$) scales play similar roles to the standard deviation ($\sigma$) and weight of the uniform distribution ($w$) in the Coherent Point Drift (CPD) method [81]: they allow us to smooth the matching in order to increase robustness to sampling artifacts.

The main difference between projection-based matching and RobOT is that the latter enforces a mass distribution constraint between the source and the target. This prevents our matching vectors from accumulating on the boundaries of the distributions of point features $(p_1, \ldots, p_N)$ and $(q_1, \ldots, q_M)$ in $\mathbb{R}^D$ [40]. This property is most desirable when the shapes to register are fully observed, with a dense sampling: as detailed in Suppl. A.5, enforcing the **global consistency** of a matching is then a worthwhile registration prior.

**Partial registration.** On the other hand, we must also stress that OT theory has known limitations [35]. First of all, the RobOT matching cannot guarantee the preservation of remarkable points or of the shapes' topologies "on its own": it should be combined with relevant feature extractors and regularizers. Going further, partial registration is not a natural fit for standard OT formulations which assume that *all* points from both shapes must be put in correspondence with each other.

To mitigate this issue, RobOT leverages the theory of *unbalanced* optimal transport [25, 105, 67, 24, 106]: we rely on *soft* Kullback-Leibler penalties to enforce a matching between the shapes A and B in Eq. (2). In practice, the RobOT confidence weights $w_i$ of Eq. (4) act as an attention mechanism: they vanish when no target feature vector $q_j$ can be found in a $\tau$-neighborhood of the source vector $p_i$ in $\mathbb{R}^D$, where $\tau$ is the *reach* scale that is associated to Eq. (2). This lets our registration method focus on reliable matches between similar features, without being fooled by strong constraints of bijectivity. As detailed in Suppl. A.3, combining standard FPFH features [100] with the rigid projection of Eq. (5) allows us to register partially observed shapes that have little overlap with each other.

# 3 Regularization and integration with a deep learning model

## 3.1 Smooth-RobOT: algebraic and optimization-based regularization

**Notations.** We now detail how to interface the weighted RobOT matching with regularization models and feature extractors that may be handcrafted [114, 101, 100] or learnt using a deep neural network [133, 46, 30, 29]. Recall that we intend to register a source point cloud $x_1, \ldots, x_N$ onto a target $y_1, \ldots, y_M$ in $\mathbb{R}^3$. Non-negative weights $\alpha_1, \ldots, \alpha_N$ and $\beta_1, \ldots \beta_M \geqslant 0$ let us take into account variations in the sampling densities and we assume that point features $p_1, \ldots, p_N, q_1, \ldots, q_M$ in $\mathbb{R}^D$ have been computed upstream by a relevant feature extractor. For every source point $x_i$, the RobOT matching layer then provides a **desired displacement** $v_i$ in $\mathbb{R}^3$ with **influence weight** $w_i \geqslant 0$.

**Smoothing in closed form.** Standard computations let us derive closed-form expressions for rigid and affine registration [57, 81, 3]. These respectively correspond to transformations:

$$\text{(Rigid RobOT)} \qquad x \in \mathbb{R}^3 \; \mapsto \; (x - x_c)\,UV^\top \qquad\qquad\qquad + x_c + v_c \in \mathbb{R}^3 \,, \quad (5)$$

$$\text{(Affine RobOT)} \qquad x \in \mathbb{R}^3 \; \mapsto \; (x - x_c)\,(\hat{X}^\top W \hat{X})^{-1}(\hat{X}^\top W \hat{Y}) + x_c + v_c \in \mathbb{R}^3 \,, \quad (6)$$

where $W = \text{Diag}(w_i) \in \mathbb{R}^{N \times N}$ is the diagonal matrix of influence weights, $x_c = \sum_i w_i x_i / \sum_i w_i \in \mathbb{R}^3$ is the barycenter of the source shape, $v_c = \sum_i w_i v_i / \sum_i w_i \in \mathbb{R}^3$ is the average desired displacement, $\hat{X} = (x_i - x_c) \in \mathbb{R}^{N \times 3}$ is the centered matrix of source positions, $\hat{Y} = (x_i + v_i - x_c - v_c) \in \mathbb{R}^{N \times 3}$ is the centered matrix of desired targets and $USV^\top$ is the singular value decomposition of $\hat{X}^\top W \hat{Y} \in \mathbb{R}^{3 \times 3}$. This corresponds to a weighted Kabsch algorithm [60]. Likewise, we implement free-form spline registration using the KeOps library [37, 20]. A Nadaraya–Watson interpolator with kernel $k : (x, y) \in \mathbb{R}^3 \times \mathbb{R}^3 \mapsto k(x, y) > 0$ [82, 124] induces a transformation:

$$\text{(Spline RobOT)} \qquad x \in \mathbb{R}^3 \; \mapsto \; x \; + \; \textstyle\sum_{i=1}^N w_i k(x_i, x) v_i \; / \; \sum_{i=1}^N w_i k(x_i, x) \in \mathbb{R}^3 \,. \quad (7)$$

**Black-box deformation models.** Going further, we interface RobOT matchings with arbitrary deformation modules $\text{Morph} : (\theta, x_i) \mapsto \hat{y}_i \in \mathbb{R}^3$ parameterized by a vector $\theta$ in $\mathbb{R}^P$. If $\text{Reg}(\theta)$ denotes a regularization penalty on the parameter $\theta$ (e.g. a squared Euclidean norm), we use standard optimizers such as L-BFGS-B [136] and Adam [63] to find the optimal deformation parameter:

$$\theta^* = \underset{\theta \in \mathbb{R}^P}{\arg\min} \;\; \text{Reg}(\theta) + \textstyle\sum_{i=1}^N w_i \| x_i + v_i - \text{Morph}(\theta, x_i) \|_{\mathbb{R}^3}^2 \,. \quad (8)$$

This optimization-based method is especially relevant in the context of computational anatomy, where smooth and invertible deformations are commonly defined through the Large Deformation Diffeomorphic Metric Mapping (LDDMM) framework [5, 8, 35]. We stress that different transformation models may result in different registration results and refer to Suppl. A.3 for further details.

Optimization-based approaches provide strong geometric guarantees on the final matching. But unfortunately, these often come at a high computational price: to register complex shapes, quasi-Newton optimizers require dozens of evaluations of the deformation model $\text{Morph}(\theta, x)$ and of its gradients. In practice, fitting a complex model to a pair of high-resolution shapes may thus take several minutes or seconds [17]. This precludes real-time processing and hinders research on advanced deformation models.

## 3.2 Deep-RobOT: registration via deep deformation prediction

**Deformation prediction.** In this context, there is growing interest in *fast* learning methods that avoid the use of iterative optimizers. The idea is to train a deep neural network $\text{Pred} : (x_i, y_j) \mapsto \theta$ that takes as input two point clouds $A = (x_1, \ldots, x_N)$, $B = (y_1, \ldots, y_M)$ and **directly predicts the optimal vector of parameters** $\theta$ for a transformation model $\text{Morph}(\theta, \cdot)$ that should map A onto B. Assuming that the prediction network Pred has been trained properly, this strategy enables real-time processing while leveraging the task-specific geometric priors that are encoded within the deformation model.

Over the last few years, numerous authors have worked in this direction for rigid registration [133, 46, 30, 2, 121, 122] and scene flow estimation [91, 126, 69, 49]. Comparable research on diffeomorphic models has focused on images that are supported on a dense grid, with successful applications to e.g. the registration of 3D brain volumes [131, 4, 107]. As of 2021, prediction-based approaches have thus become standard methods for 3D shape registration.

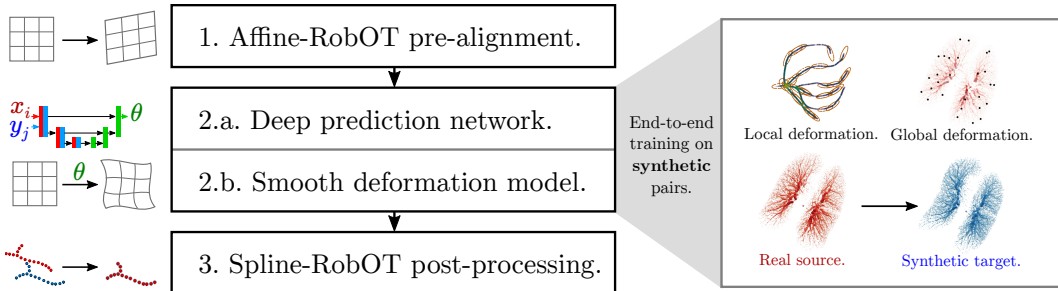

Figure 2: **The D-RobOT architecture. Left:** We apply three successive registration modules that bring the moving source shape increasingly close to the fixed target point cloud. On the one hand, the RobOT-based pre-alignment (1) and fine-tuning (3) steps take the $(x, y, z)$ coordinates as input features and do not require any training. On the other hand, our deep registration module (2) relies on a multi-scale point neural network "Pred : $(x_i, y_j) \mapsto \theta$" (2.a) and a task-specific deformation model "Morph$(\theta, x_i) \mapsto \hat{y}_i$" (2.b). We train it end-to-end on a dataset of synthetic pairs of shapes with known ground truth correspondences. **Right:** To generate these pairs, we apply random deformations to real source shapes. For lung registration, we apply successively a vessel-preserving local perturbation and a smooth global deformation – as detailed in Suppl. A.2.

**The D-RobOT model.** In practice though, prediction methods still face three major challenges:

1. Common architectures may not be equivariant to **rigid or affine transformations**.
2. Dense 3D annotation is expensive, especially in a medical context. As a consequence, most predictors are trained on **synthetic data** and have to overcome a sizeable **domain gap**.
3. Predicted registrations may be **less accurate** than optimization-based solutions. Training registration networks to pixel-perfect accuracy is notoriously hard, with diminishing returns in terms of model size and number of training samples.

We propose to address these issues using RobOT layers for pre-alignment and post-processing. Our Deep RobOT (D-RobOT) model is an end-to-end deep learning architecture that is made up of three consecutive steps that we illustrate in Figure 2 and detail in Suppl. A.4:

1. **OT-based pre-alignment.** We use the rigid or affine S-RobOT models of Eqs. (5-6) to normalize the pose of the source shape. This is a fast and differentiable pre-processing.
2. **Deep registration module.** We combine a deep predictor with a task-specific deformation model to register the pre-aligned source onto the target. For the prediction network Pred : $(x_i, y_j) \mapsto \theta$, we use a multiscale point neural network that is adapted from the PointPWC-Net architecture [126]. We refer to Suppl. A.4 for a full description of our architecture and training loss.
3. **OT-based post-processing.** In order to reach "pixel-perfect" accuracy, we use the spline S-RobOT deformation model of Eq. (7) with a task-specific kernel $k(x, y)$.

**Complementary strengths and weaknesses.** We apply these three steps successively, which brings the moving source A $= (x_1, \ldots, x_N)$ increasingly close to the fixed target B $= (y_1, \ldots, y_M)$. Remarkably, each step of our method covers for the weaknesses of the other modules: the RobOT-based **pre-alignment** makes our pipeline robust to changes of the 3D acquisition parameters; our multiscale neural **predictor** is able to match corresponding key points quickly, even in complex situations; the domain-specific **deformation model** acts as a regularizer and improves the generalization properties of the deep registration module; the RobOT-based **fine-tuning** improves accuracy and helps our model to overcome the domain gap between synthetic and real shape data.

As detailed below, the D-RobOT model generalizes well outside of its training dataset and outperforms state-of-the-art methods on several challenging problems. We see it as a pragmatic architecture for shape registration, which is easy to deploy and tailor to domain-specific requirements. As discussed in Suppl. A.4, we found that introducing sensible geometric priors through our RobOT layers and the deformation model Morph : $(\theta, x_i) \mapsto \hat{y}_i$ results in a "forgiving" pipeline: **our model produces accurate results, even when trained on synthetic data that is not very realistic**. In a context where generating plausible 3D deformations is easier than developing custom registration models for every single task (e.g. in computational anatomy), we believe that this is an important observation.

# 4   Scene flow estimation

**Benchmark.** We now evaluate our method on a standard registration task in computer vision: the estimation of scene flow between two successive views of the same 3D scene. We follow the same experimental setting as in [126, 49], with full details provided in Suppl. A.4.3:

1. We train on the synthetic **Flying3D** dataset [75], which is made up of multiple moving objects that are sampled at random from ShapeNet. We take 19,640 pairs of point clouds for training, with dense ground truth correspondences.
2. We evaluate on 142 scene pairs from **Kitti**, a real-world dataset [78, 77]. We conduct experiments using 8,192 and 30k points per scan, sampled at random from the original data.

**Performance metrics.** We evaluate all methods as in [126]: **EPE3D** is the average 3D error, in centimeters; **Acc3DS** is the percentage of points with 3D error $< 5$ cm or relative error $< 5\%$; **Acc3DR** is the percentage of points with 3D error $< 10$ cm or relative error $< 10\%$; **Outliers3D** is the percentage of points with 3D error $> 30$ cm or relative error $> 10\%$; **EPE2D** is the average 2D error obtained by projecting the point clouds onto the image plane, measured in pixels; **Acc2D** is the percentage of points with 2D error $< 3$ px or relative error $< 5\%$. As detailed in Suppl. A.6, all run times were measured on a single GPU (24GB NVIDIA Quadro RTX 6000).

**Methods.** We study a wide range of methods and report the relevant metrics in Fig. 4:
In the **upper third** of the table, we report results for unsupervised methods that do not require ground truth correspondences for training. This includes the "raw" RobOT matching of Eq. (4), computed on $(x, y, z)$ coordinates in $\mathbb{R}^3$ with a *blur* scale $\sigma = 1$ cm and a *reach* scale $\tau = +\infty$. Please also note that PWC refers to an improved version of PointPWC-Net, released on GitHub (`https://github.com/DylanWusee/PointPWC`) after the publication of [126] with a self-supervised loss.
In the **central third** of the table, we benchmark a collection of state-of-the-art point neural networks.
In the **lower third** of the table, we study the influence of our RobOT-based layers. The methods "Pre + FLOT/PWC + Post" correspond to the FLOT and PointPWC-Net architectures, with the additional pre-alignment and post-processing modules of Sec. 3.2. The last two lines correspond to the full D-RobOT architecture (with a spline deformation model) whose training is detailed in Suppl. A.4.3.

**Results.** We make three major observations:

1. Without any regularization or training, a simple RobOT matching on high-resolution data outperforms many deep learning methods in terms of speed, memory footprint and accuracy (line 6 of the table). This surprising result is strong evidence that **geometric methods and baselines deserve more attention** from the computer vision community.
2. In the lower third of the table, RobOT-enhanced methods **consistently outperform state-of-the-art methods** by a wide margin.
3. As shown in Fig. 3, these improvements are most significant on **high-resolution data**.

Overall, as detailed in Suppl. A.4.3 and A.5, we observe that optimal transport theory is especially well suited to scene flow estimation. Assuming that ground points have been removed from the 3D frames (as a standard pre-processing), most object displacements can be explained as translations and small rotations: this is an ideal setting for our robust geometric method.

# 5   Registration of high-resolution lung vessel trees

**PVT1010: a new dataset for lung registration.** Going further, we introduce a new dataset of 1,010 pairs of pulmonary vessel trees that must be registered between expiration (source) and inspiration (target). Due to the intricate geometry of the lung vasculature and the complexity of the breathing motion, the registration of these shapes is a real challenge.

As detailed in Suppl. A.1, we encode our $1,010 \times 2$ vessel trees as high-resolution 3D point clouds ($N = M = 60$k points per tree). For each point, we also provide a local estimate of the vessel radius that we use as an additional point feature or as a weight $\alpha_i$ or $\beta_j$ in Eq. (2). Our first 1,000 pairs of 3D point clouds are provided without ground truth correspondences; for all of our experiments, we randomly sample 600 training and 100 validation cases from this large collection of unannotated patients. The last 10 cases correspond to the 10 DirLab COPDGene pairs [19]: they come with 300 expert-annotated 3D landmarks per lung pair, that we use to test our methods.

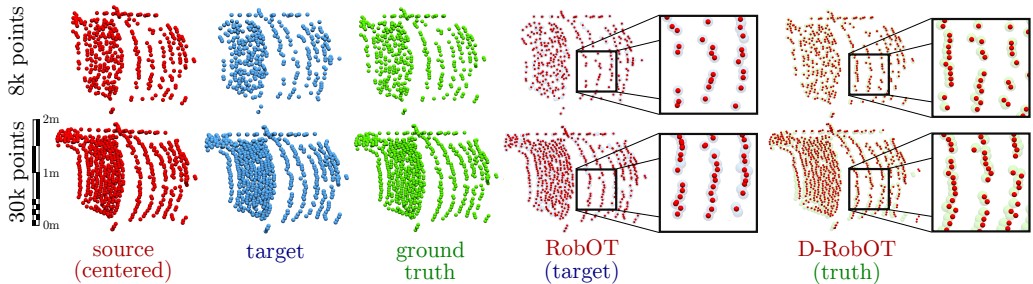

Figure 3: **Influence of the sampling density** on a detail (top car) of the Kitti frames of Fig. 1.b. **First row:** On sub-sampled 3D scenes, the regularizing priors of the D-RobOT architecture prevent over-fitting to the random sampling patterns of the target point cloud (blue). The D-RobOT output (last column, red) is very close to the ground truth scene flow (green). **Second row:** Increasing the number of points per frame reduces the influence of sampling artifacts. The simple RobOT baseline (fourth column, red) still over-fits to the target (blue) but becomes remarkably competitive.

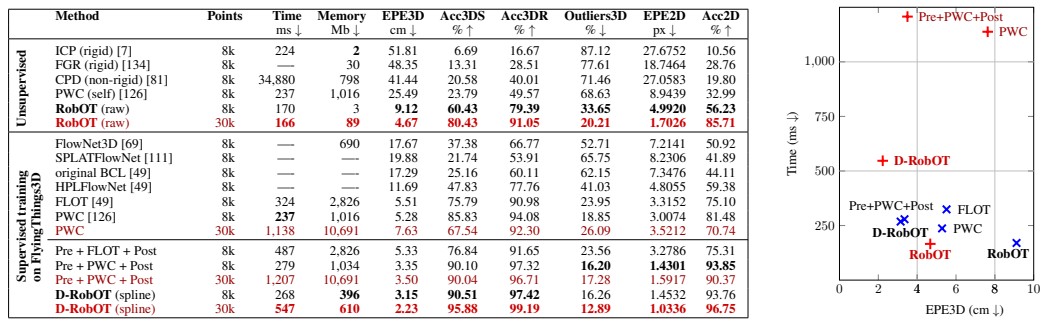

| | Method | Points | Time ms↓ | Memory Mb↓ | EPE3D cm↓ | Acc3DS %↑ | Acc3DR %↑ | Outliers3D %↓ | EPE2D px↓ | Acc2D %↑ |
|---|---|---|---|---|---|---|---|---|---|---|
| **Unsupervised** | ICP (rigid) [7] | 8k | 224 | **2** | 51.81 | 6.69 | 16.67 | 87.12 | 27.6752 | 10.56 |
| | FGR (rigid) [134] | 8k | — | 30 | 48.35 | 13.31 | 28.51 | 77.61 | 18.7464 | 28.76 |
| | CPD (non-rigid) [81] | 8k | 34,880 | 798 | 41.44 | 20.58 | 40.01 | 71.46 | 27.0583 | 19.80 |
| | PWC (self) [126] | 8k | 237 | 1,016 | 25.49 | 23.79 | 49.57 | 68.63 | 8.9439 | 32.99 |
| | **RobOT (raw)** | 8k | 170 | 3 | **9.12** | **60.43** | **79.39** | **33.65** | **4.9920** | **56.23** |
| | RobOT (raw) | 30k | 166 | 89 | 4.67 | 80.43 | 91.05 | 20.21 | 1.7026 | 85.71 |
| **Supervised training on FlyingThings3D** | FlowNet3D [69] | 8k | — | 690 | 17.67 | 37.38 | 66.77 | 52.71 | 7.2141 | 50.92 |
| | SPLATFlowNet [111] | 8k | — | — | 19.88 | 21.74 | 53.91 | 65.75 | 8.2306 | 41.89 |
| | original BCL [49] | 8k | — | — | 17.29 | 25.16 | 60.11 | 62.15 | 7.3476 | 44.11 |
| | HPLFlowNet [49] | 8k | — | — | 11.69 | 47.83 | 77.76 | 41.03 | 4.8055 | 59.38 |
| | FLOT [49] | 8k | 324 | 2,826 | 5.51 | 75.79 | 90.98 | 23.95 | 3.3152 | 75.10 |
| | PWC [126] | 8k | 237 | 1,016 | 5.28 | 85.83 | 94.08 | 18.85 | 3.0074 | 81.48 |
| | PWC | 30k | 1,138 | 10,691 | 7.63 | 67.54 | 92.30 | 26.09 | 3.5212 | 70.74 |
| | Pre + FLOT + Post | 8k | 487 | 2,826 | 5.33 | 76.84 | 91.65 | 23.56 | 3.2786 | 75.31 |
| | Pre + PWC + Post | 8k | 279 | 1,034 | 3.35 | 90.10 | 97.32 | **16.20** | **1.4301** | **93.85** |
| | Pre + PWC + Post | 30k | 1,207 | 10,691 | 3.50 | 90.04 | 96.71 | 17.28 | 1.5917 | 90.37 |
| | **D-RobOT (spline)** | 8k | 268 | 396 | **3.15** | **90.51** | **97.42** | 16.26 | 1.4532 | 93.76 |
| | **D-RobOT (spline)** | 30k | 547 | 610 | **2.23** | **95.88** | **99.19** | **12.89** | **1.0336** | **96.75** |

Figure 4: **Evaluation on the Kitti dataset for 3D scene flow.** Black numbers and crosses (×) correspond to results on scene pairs that are sampled with 8,192 points per frame; red numbers and plus signs (+) correspond to scene pairs that are sampled with 30,000 points per frame.

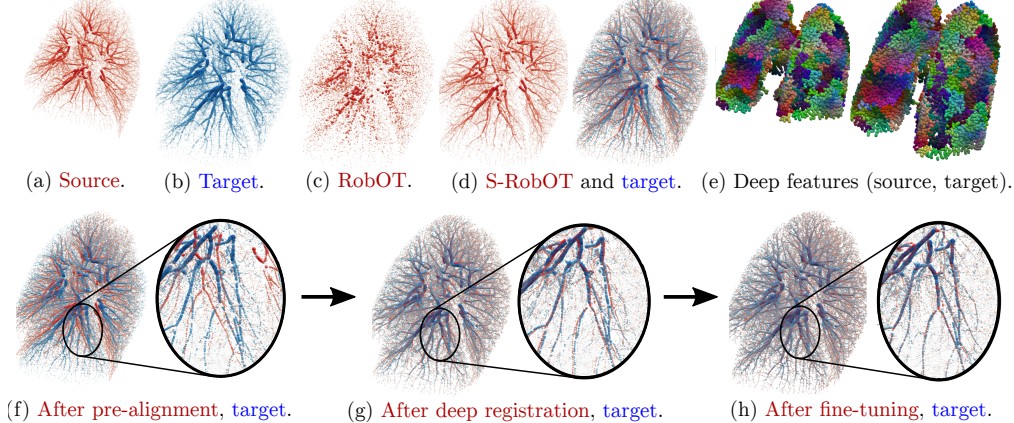

(a) Source.   (b) Target.   (c) RobOT.   (d) S-RobOT and target.   (e) Deep features (source, target).

(f) After pre-alignment, target.   (g) After deep registration, target.   (h) After fine-tuning, target.

Figure 5: **Registration of lung vascular trees. First row: S-RobOT** registration with the deep features of Suppl. A.3.3. We display: (a) the source and (b) target shapes; (c) the "raw" RobOT registration; (d) a smoother S-RobOT registration with spline regularization, an overlap of this result with the target shape; (e) a visualization of the deep features on a pair of lung vascular trees, with colors that correspond to a t-SNE embedding of the point features in color space [116]. **Second row: D-RobOT** registration. We display the successive steps of our model: (f) pre-alignment with affine S-RobOT; (g) deep registration with an LDDMM model; (h) fine-tuning with spline S-RobOT.

Table 1: **3D registration errors on the 3,000 expert-annotated DirLab landmarks.** The † symbol denotes methods that are based on *image keypoints* [50, 51] and are evaluated on the original DirLab *image* dataset; all other approaches are tested on our *point clouds*. Due to its large memory requirements, CPD is tested on clouds of 20k points (instead of 60k).

|  | Method | Average error mm ↓ | Percentiles 25% mm ↓ | 50% mm ↓ | 75% mm ↓ | Time s ↓ |
|---|---|---|---|---|---|---|
| No training | Input data | 23.30 | 13.18 | 22.22 | 31.65 | — |
| | ICP (affine) [7] | 15.05 | 9.60 | 14.06 | 20.01 | 0.52 |
| | CPD (non-rigid) [81] | 9.30 | 5.95 | 8.60 | 11.83 | 332.60 |
| | RobOT (affine) | 10.45 | 6.01 | 9.83 | 13.97 | 0.18 |
| | RobOT (raw) | 9.41 | 4.89 | 8.35 | 13.04 | **0.15** |
| Supervised | DGCNN-CPD† [50] | 4.30 | — | — | — | — |
| | DispEmd† [51] | 3.42 | — | — | — | — |
| | S-RobOT (spline) | 5.72 | 3.19 | 5.04 | 7.35 | 2.77 |
| | S-RobOT (LDDMM) | 5.48 | 2.86 | 4.44 | 7.14 | 42.30 |
| | D-RobOT (raw) | 3.40 | 1.40 | 2.58 | 3.69 | 1.26 |
| | D-RobOT (spline) | 2.95 | 1.30 | 2.50 | 3.19 | 1.87 |
| | D-RobOT (LDDMM) | **2.86** | **1.25** | **2.23** | **3.11** | 1.92 |

**Data augmentation.** The imbalance between our large training set and the small collection of 10 test cases reveals a fundamental challenge in computational anatomy: annotating pairs of 3D medical shapes with dense correspondences is prohibitively expensive. To work around this problem, we train our networks on synthetic deformations of the $600 \times 2$ lung shapes that make up our training set. As detailed in Suppl. A.2, we use a two-scales random field to generate a wide variety of deformations. This allows us to create a suitable training set with dense "ground truth" correspondences.

Overall, as discussed in Suppl. A.4.2, we found that supervised training on synthetic deformations is both easier and more efficient than unsupervised training on real lung pairs. From the chamfer and Wasserstein distances [35] to local Laplacian penalties [126], none of the unsupervised loss functions that we experimented with was able to deal with the complex geometry of our lung vascular trees.

**Methods.** We benchmark a wide range of methods in Fig. 5 and Tab. 1. In the **upper half** of the table, we evaluate geometric approaches that require no training; in the **lower half** of the table, we benchmark scalable point neural networks that we trained on our synthetic dataset. We evaluate three types of RobOT-based approaches: a simple RobOT matching computed using $(x, y, z)$ coordinates as point features, that may either be "raw" as in Eq. (4) or regularized using Eq. (6); an S-RobOT matching that we compute using the deep features of Suppl. A.3.3 and regularize with the spline smoothing of Eq. (7) or the LDDMM optimization of Eq. (10); a D-RobOT architecture that we pair with three deformation models $\text{Morph}(\theta, x_i) \mapsto \hat{y}_i$ and describe in depth in Suppl. A.4.2.

**Results.** We provide additional experiments in Suppl. A.4.2 and make three major observations:

1. **Explicit regularization** with a spline or LDDMM deformation model is key. Model-free architectures that predict raw 3D correspondences produce non-smooth results that are not anatomically plausible, even when they are trained entirely on smooth deformations.
2. The D-RobOT architecture combines a **high acccuracy** with fast run times.
3. Most remaining errors occur at the **boundary of the lungs**, where acquisition artifacts prevent the thinnest vessels from being sampled reliably in our point cloud representation.

## 6   Conclusion, limitations and future work

Our work builds upon a decade of active research in the field of computational optimal transport. We leverage major advances on RobOT solvers to define a new matching layer which is a plug-and-play replacement for nearest neighbor projection. This operation has two major uses in 3D shape registration: first, it provides a **very strong geometric baseline** for e.g. scene flow estimation; second, it increases the accuracy and generalization abilities of point neural networks on finely sampled 3D shapes. We see D-RobOT as a **mature and versatile architecture** for shape registration which is easy to train and adapt to task-specific requirements in e.g. medical imaging.

Going forward, we see three main ways of improving this work. First, we still have to investigate in depth the important problem of occlusions and **partial acquisitions**. Second, integrating **task-specific features** beyond the $(x, y, z)$ point coordinates is often key to perfect results. In the specific setting of lung registration, working with image-based features or focusing on branching points could be a way of improving performance at the cost of portability: recent works such as [52, 53] are an excellent source of inspiration. Finally, we believe that high-quality **software packaging** is an important part of research in our field. We intend to keep working on the topic and distribute our methods through a user-friendly Python library for widespread use by the scientific community.

## Acknowledgments and Disclosure of Funding

Research reported in this publication was supported by the National Heart, Lung, and Blood Institute of the National Institutes of Health under award numbers R01HL149877 and R01HL116473. The content is solely the responsibility of the authors and does not necessarily represent the official views of the National Institutes of Health. The authors would also like to thank the anonymous reviewers for their most valuable advice.

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
