# OpenReview forum: "Accurate Point Cloud Registration with Robust Optimal Transport"
_NeurIPS.cc/2021/Conference — NeurIPS 2021 Poster_

### Official Review · Reviewer_Yt1x · 2021-07-11

**Rating:** 6
**Confidence:** 3

**Summary:**

The paper proposes a computationally efficient point cloud registration method based on Optimal Transport. The DRobOT pipeline relies on a deep module that predicts parameters of a registration model, plus pre and post processing steps based on RobOT. The paper shows results for scene flow and point cloud lung registration.

**Ethical Concerns:**

I do not see any ethical concern in this work.

**Limitations And Societal Impact:**

I do not see any potential negative societal impact in this work.

**Main Review:**

PROS
======
EFFICENCY\
The method shows efficiency in terms of computation without an explosion of memory. This is a relevant aspect that has a high application interest in several fields. The method specifies that code and data will be available, and these are both remarkable contributions.

REAL-WORLD RESULTS\
I like that the shown results are on real data, with an extensive comparison with many different methods on two different applicative domains. It is also providing an ablation on different aspects of the method; I think this is a good validation for the method.

DIFFERENT REGISTRATION MODELS\
The method admits different kinds of registration models, which may be properly selected for different applicative purposes. Also, the method can face partiality (as shown in a toy example from supplementary material), which I consider an interesting and challenging setting.

CONS
======
PRESENTATION\
I think the paper is a bit verbose, and the extensive use of paragraphs make it hard to understand the structure of the underlying story. For example, Section 3 is a bit confused; the first paragraphs are about derivations useful for the method, then some properties are listed (partial registration, black-box deformation models), to move to lung data presentation and then experimental setting. I would suggest rethinking Section 3 and 4 to better group a) elements of the method (and maybe an enumerate workflow of different steps), b) analysis of the method properties, c) involved datasets, d) experiments on the two different scenarios. However, I would suggest revising the text and finding ways to make it more compact and schematic, and also adding some support Figures to improve the narration (e.g., some toy examples of the process, like the sphere to cube from the supplementary). Consequently, I would suggest rethinking page 7, since it breaks the text and the reader has some work to do to go back and forth between the pages. I would suggest highlighting in bold the first and the second best values in Figure 3, 4 and Table 1, to ease the inspection of the values. Understanding the lunge images is a bit complicated for non-expert users (like me) in particular when they are aligned; maybe highlighting errors in visualization (e.g., heat map) would be beneficial. Also, I think that some image examples of KITTI dataset could be moved from supplementary to the main manuscript. I would suggest including losses into the main manuscript and removing Algorithm 1, which seems not significantly informative. Finally, the title is a bit generic; I would suggest adding something that emphasizes the specific contribution of the paper (e.g., the use of a deep network, the mention of the efficiency, the use of the DRobOT, ...).

FORMULATION VS IMPLEMENTATION \
In several parts, the problem formulation is somehow obfuscated by implementation aspects. For example, in the "The weighted RobOT matching" paragraph, the formulation is mixed with the use of libraries (GeomLoss, PyTorch, KeOps, NumPy). It is not completely clear to me if the efficiency comes from the smart formulation or from the library implementations. I would like to read a comment on this aspect; in general, I would suggest creating a "implementation" subsection where list all the aspects of the coding.

SUPERVISION \
While the method presents general characteristics in the deformation paradigm, it requires a supervised setting for training. I think this is somehow limiting, considering that Optimal Transport finds application in many fields in which a ground truth correspondence is, in general, an ill-posed problem or hard to obtain.

MINOR COMMENTS AND QUESTIONS
=======
a) I see from Figure 5 of Supplementary Materials that the method can also solve for partiality. Do you have any insight on the amount of partiality that the method could tackle? I think an evaluation on this aspect would also make the paper significantly stronger. \
b) Do you see any limitations to use more complex deformation paradigms, e.g., the parameters of morphable models (which may also be some hundreds)?

RATING JUSTIFICATION
=========
The idea is not particularly surprising, but the evaluation and efficiency analysis are interesting, and the method seems to be of potential interest for NeurIPS community. I would suggest revising the presentation and work on the clarity of the method presentation.

**Time Spent Reviewing:**

5

---

> ### Author Response · Authors · 2021-08-10
> **Answer to Reviewer 4**
>
> We thank the reviewer for the detailed comments, constructive advice and positive feedback. We now address the main comments, summarized below:
>
>
> ### Presentation
>
> We fully agree with the reviewer and appreciate the very helpful comments.
>
> To be honest, we found it difficult to include all of the necessary details on OT theory and our extensive experiments in the 9-page submission. As a consequence, we understand that our submission ended up being fairly "compact" and would benefit from a bit more breathing space.
>
> In the final version of the paper, we will fix this issue by **taking advantage of the extra page** that is allowed at the end of the review process. Most importantly, we will follow the reviewers' suggestions as follows:
>
> - Instead of keeping two separate presentations for RobOT-P and D-RobOT, we will **re-organize Sections 3 and 4 as suggested** (3.A - Methods and algorithms; 3.B - Main properties; 3.C - Implementation; 4.A - Dataset and results for Flying3D/Kitti; 4.B - New dataset and results for the lung registration task). Page 7 would then fit nicely within Section 4.B.
> - Replace Algorithm 1 by explicit equations for the spline and LDDMM loss functions.
> - Move back Figure 6 and our paragraphs on limitations and social impact (+ acknowledgements for the reviewers' help and funding agencies) from the Supplementary Materials to the main paper.
> - Add a qualitative illustration on Kitti to the main paper.
> - Re-draw all of the lung figures with a **cleaner 3D graphic theme**. Notably, we found that using opaque points with varying radii (colored in red/blue for the moving/target shapes, respectively) produces pictures that are much easier to read than the blurry, opacity-based visualizations of our submission. In the Supplementary Materials, we will also include results that display the final deformation as a quiver-like plot, colored by the magnitude of the error made on the nearest annotated landmark.
> - Highlight the most relevant values in our tables.
>
>
> ### Formulation vs implementation
>
> The numerical efficiency of our method results from its **close fit to the strengths and weaknesses of GPU hardware:** it avoids non-contiguous memory accesses and the storage of large memory buffers. Notably, we decided to focus on the formulation of Eqs. (3-4) because it is a **perfect fit for the KeOps library,** thus streaming much faster on GPUs than common alternatives. Among the many possible choices for the encoding of the unbalanced transport plan (using e.g. a dense array or a sparse matrix), our "weighted Monge map" formulation is the one that is best suited to a fast and scalable implementation: we refer to our answer to Reviewer 2 (qUAf) for details on this point.
>
> We also believe that the simplicity of our formulation will play a key role in its widespread adoption outside of the OT literature. The fact that our weighted RobOT matching can be used as a **plug-in replacement for nearest neighbor projection** makes it especially appealing to the wider community in computer vision.
>
> As detailed above, we will follow the reviewer's advice and discuss numerical aspects in a dedicated subsection.
>
>
> ### Supervision
>
> We agree that this question is important: as detailed in our answer to Reviewer 2 (qUAf), annotated data can be a scarce commodity.
>
> In this context, we would like to stress that our models for scene flow estimation and lung pair registration are **entirely trained on synthetic data:**
>
> - Our results on Kitti are achieved after a training on the synthetic Flying3D dataset.
> - Our results on Dirlab are achieved after a training on synthetic deformations of our (real but unannotated) lung vessel trees.
>
> In other words, we show that simulating deformations on an (unsupervised) dataset of input shapes is a perfectly viable strategy. We reach state-of-the-art accuracy on two very challenging registration tasks.
>
>
> We understand that this answer may seem underwhelming to specialists of OT theory. In light of recent progresses in the field, we would also have loved to see a "fully unsupervised" OT loss produce state-of-the-art performance with no need for a domain-specific generator of synthetic data. This was our first approach -- which allowed us to obtain decent results.
> Nevertheless, as discussed in Section A.4.2 of the Supplementary Materials, we found that the "data augmentation" method with synthetic deformations is **significantly more robust** and easy to train in practice, with a **higher ceiling** in terms of accuracy.
>
> These results make sense in a context where **generating realistic 3D deformations is easier than improving upon the well-known limitations of OT theory** (especially with respect to the preservation of topology). We thus chose to report them honestly, without over-stating the accuracy of pure OT tools on real data. In light of our experiments, we strongly believe that combining powerful mathematical tools with domain-specific regularization methods is the most reliable way of designing accurate shape registration methods.
>
> Since these experimental results and our conclusions are likely to be of interest to the community, we will make this message more explicit in the final version of our paper.
>
>
>
>
> ### Minor comments:
> **a) What is the amount of partiality that the method could tackle?**
>
> This is very good question. From what we observe, results on partial registration largely depend on the quality of the input features ($p_i$, $q_j$ in Eq. (2)). If the features are representative and discriminate well between different parts of the input shapes, unbalanced OT theory provides a nice mechanism to weigh the confidence of the RobOT matching.
>
> In practice, for rigid transformations with only six parameters to estimate (as illustrated in Fig. 5) the small size of the overlap region doesn't really matter as long as the confidence map masks region correctly. We found it to be the case with the relatively simple FPFH features.
>
> On the other hand, for more flexible non-rigid registration, the problem becomes harder to solve as sensitivity to the quality of the confidence map increases. As discussed in our answer to Reviewer 1 (Cd96), this is a question that we will investigate in future work.
>
> **b) Do you see any limitations to use more complex deformation paradigms, e.g. morphable models that may have hundreds of parameters?**
>
> Our framework is fully compatible with complex deformation models. We stress that our LDDMM deformations are "non-parametric" (i.e. are parameterized by a dense vector field with thousands of degrees of freedom, the "shooting momentum") and are more complex than a majority of Morphable Models in computer vision.
>
> Notably, 3D Morphable Models and GP-Morphable Models fit perfectly within our "RobOT-P" framework. We will add code examples for these models in our reference implementation -- alongside the rigid, affine, spline and LDDMM models that have already been discussed in the paper.

---

> > ### Comment · Reviewer_Yt1x · 2021-08-31
> > **Final Comment**
> >
> > Dear Authors,
> > Thank you for your effort; I would like to acknowledge that I have appreciated your replies.
> >
> > Across the reviews, I see there is a general positive feeling about this work, mainly due to:
> >
> > - Experiments: the work shows convincing results, and it improves state of the art in non-trivial settings (e.g., 60K points registration)
> > - Applicability: the method seems efficient and simple; these points could attract wide interest from the community
> >
> > On the other hand, the concerns are:
> > - Novelty\Originality: some of us consider the proposed method to connect existing techniques\libraries without any particular new insight.
> > - Partiality: there is a lack of testings in the partial setting, but the rebuttal promised more results for the final version of the paper.
> >
> > Given so, my feeling on this paper is positive, and I am favourable to recommend acceptance, suggesting for the final version to include experiments on partiality, 3D\GP-Morphable Models tests, limitations discussion and fixing all the presentations issues highlighted.
> >
> > I wish you good luck with your work!

---

### Official Review · Reviewer_Y4k6 · 2021-07-16

**Rating:** 5
**Confidence:** 4

**Summary:**

This paper proposes a transport optimal method to calculate the assignment by adding several regularizers.


**Main Review:**

Originality: This paper adds several regularizers to the standard optimal transport method.

Quality: Firstly, the contribution is limited. Secondly, I am a bit worried about whether the better performance comes from the regularizers since the proposed method has pre-alignment and post-processing. The key contribution of this paper lies in post-processing. How about the performance by replacing the post-processing with standard transport optimal method. This research work of this submission seems not complete. I recommend the authors conduct these experiments.

Clarity: The overall writing is well and very easy to follow.

significance: The results are important but not convince to me. More ablation study is still required to demonstrate the value of this work.

**Time Spent Reviewing:**

7

---

> ### Author Response · Authors · 2021-08-10
> **Answer to Reviewer 3**
>
> We thank the reviewer for the time and positive comments on the clarity and significance of our work. We summarize and address the main comments below.
>
> ### The contribution is limited.
>
>
> We respectfully disagree. As detailed in Lines 92-109 of the paper, our main contributions read as follows:
>
> - We **highlight our formulation of Eqs. (3-4) as a simple, effective and easy-to-deploy product of modern OT theory for 3D shape analysis.** In light of the widening gap between cutting edge OT theory and common practice in computer vision (where most authors still rely on the Auction algorithm and the vanilla Sinkhorn solver), we believe that this is an important scientific contribution.
> - We demonstrate that Robust OT is relevant to tackle a wide range of registration problems. We validate this claim by studying both optimization-based and deep-learning methods for point cloud registration. To the best of our knowledge, our paper provides the **most comprehensive study of Optimal Transport tools on real 3D shape data**.
> - We release a **challenging new dataset of lung vessel trees sampled at a high resolution** for research in 3D shape analysis.
>
>
> Most importantly, we fully **investigate the interaction of an advanced mathematical theory** (Robust Optimal Transport) with **state-of-the-art methods** in 3D shape analysis (from spline and diffeomorphic models to deep neural networks).
>
> From an experimental perspective, our most remarkable results are that:
>
>
> - Even our *simplest* (optimization-based) RobOT method outperforms most concurrent deep learning approaches in terms of accuracy, speed and memory on the popular Kitti dataset. This is **strong evidence that modern geometric tools** (such as Robust OT theory) **deserve more attention from the computer vision community**.
> - Our proposed D-RobOT architecture can be trained entirely on synthetic data and still **generalizes well** to the unseen, complex shapes and deformations of the **Kitti** and **COPDGene DirLab** datasets -- as detailed in our answer to Reviewer 2 (qUAf).
> - Our end-to-end D-RobOT model combines an accurate and robust OT solver with an expressive deep learning architecture to achieve **state-of-the-art performance on two challenging datasets**.
>
> Overall, our investigation combines an in-depth methodological work with a rigorous experimental study: our paper **bridges the gap between cutting edge OT theory and real-life applications.**
> We thus believe that it will be of significant interest to the optimal transport and computer vision communities at NeurIPS 2021.
>
>
> ### Does the better performance come from the regularizers instead of the RobOT matching, since the proposed method has pre-alignment and post-processing? The key contribution of this paper lies in post-processing: I suggest an ablation study where the post-processing is replaced by a standard optimal transport method.
>
>
> We stress that **our paper already contains results for "pure" optimal transport methods** -- we will make this point fully explicit in the final version of the paper.
>
> In Figures 2, 3 and 4, the "RobOT", "RobOT (non-param)" and "RobOT (dense)" legends all refer to results that do not go through our optimization- or deep-learning based regularization modules (labeled as RobOTP and DRobOT, respectively). In other words, these results are computed using nothing but (modern, unbalanced) OT theory on 3D point clouds with xyz features.
>
> In practice, we observe that these methods can achieve **good  results in terms of accuracy** (at a very small computational cost), but **cannot recover plausible deformations that are both large and complex**.
> This is most evident in the 2nd picture on the first line of Figure 2: the "pure RobOT" final registration result is not satisfying, since the lung vessels are scattered to pieces by the (un-regularized) OT matching.
>
> A key message of our paper is therefore that **in most real-life scenarios, "sharp" OT matchings should be regularized** to produce plausible deformations. Sections 3 and 4 of our paper describe how to perform such a regularization within the two most popular frameworks in 3D shape analysis: "optimization-based" and "end-to-end deep learning" registration.
>
> Overall, our results show that combining a RobOT matching with a domain-specific regularization strategy is a simple way of **getting the best of both worlds**: we reach **state-of-the-art** performance on a wide range of registration problems, with models that generalize well to **unseen data**.
>
>
>
> ###  More ablation study is still required to demonstrate the value of this work.
>
> We thank the reviewer for this honest feedback: we agree that ablation studies are key to convince readers of the interest of our method.
>
> As detailed above, our paper already contains several experiments that study the **influence of the regularization modules** on the quality of the final RobOT matching.
>
> Going further, we will also add an extra ablation study that **compares our OT-based postprocessing** ("fine-tuning") method with **simpler approaches that rely on K-Nearest Neighbors projections**.
> In practice, for the lung registration task, we observe that our proposed post-processing (RobOT + vessel-aware anisotropic smoother) achieves the best performance compared with RobOT only, KNN only and KNN + anisotropic smoother.
>
> This result will be a welcome addition to the **extensive collection of ablation studies that is already present in the main paper:**
> - Figures 3 and 4 document results with a **wide range of regularizers** (rigid, non-rigid, spline, LDDMM, etc.). These are compared with a representative collection of **prior works**. **Qualitative visualizations** are displayed in Figures 2, 6 and 9, both on toy data and on complex lung vessel trees.
> - Table 1 provides a detailed ablation study for the **neural architecture** of our best-performing method DRobOT (LDDMM).
>
> And in the **Supplementary Materials:**
> - Table 2 provides an ablation study on the usefulness of the **OT-based postprocessing step** for lung registration.
> - Table 3 provides an ablation study on the impact of the **point sampling density** for the Kitti dataset.
>
> Combined with the detailed discussion of Section 2, we thus believe that our paper makes a very strong case for the widespread use of Robust OT in 3D shape registration.

---

### Official Review · Reviewer_qUAf · 2021-07-16

**Rating:** 6
**Confidence:** 4

**Summary:**

This paper discusses the application of optimal transport theory in and contributed several useful practices to shape matching. First, the proposed RobOT method implements the traditional entropy-regularized unbalanced optimal transport using the kernel method and reduces the memory footprint to linear, enabling the processing of over 60k points. Second, the paper introduces several effective strategies to regularize/project the transport vectors given a deformation model (RobOTP). Third, the deep deformation framework (DRobOT) is proposed by sandwiching the parameter prediction module with pre-alignment and post-optimization steps, which lead to SoTA performance on a new lung dataset and scene flow datasets. Apart from the algorithmic side, the authors also give a clear summary which cover most of the vision algorithms using optimal transport.

**Limitations And Societal Impact:**

The paper misses a broader impact statement and the main paper fails to discuss the limitations - these are rather relegated to the supplementary material. I would recommend that the paper includes a brief discussion in the main paper.

**Main Review:**

PROs
- The paper provides a good summary of the usage of optimal transport in computer vision pipelines. The analysis is thorough and clear.
- The experiments on Lung registration and scene flow estimation show promising results compared to the baselines.
- The proposed method can scale up to 60k points for registration tasks in a deep OT context, which is a non-trivial advancement.

CONs
- One of the core contributions of the paper, i.e., RobOT, seems to be a programming/engineering trick and a simple stacking of GeomLoss and KeOps library (which is btw already included in GeomLoss several years ago). Please refer to Line155-Line164 for this claim. Although the kernel trick can significantly reduce the memory footprint (i.e. from quadratic to linear), no extra computational effort is saved and we lost the efficiency provided by the cublas libraries by removing matrix multiplication. So I'm curious how and why is the proposed method faster and more robust?

- The size of the test set for lung registration is too small (only 10 compared to the 800-200 train/val split used for training). I would expect the paper to re-split their 1k-sized annotated dataset and used at least 100 samples for testing.

- The following works are certainly relevant and are worth comparing to (given that they share a similar philosophy to this paper):
Gojcic, Zan, et al. "Weakly Supervised Learning of Rigid 3D Scene Flow." Proceedings of the IEEE/CVF Conference on Computer Vision and Pattern Recognition. 2021.
Li, Ruibo, et al. "HCRF-Flow: Scene Flow from Point Clouds with Continuous High-order CRFs and Position-aware Flow Embedding." Proceedings of the IEEE/CVF Conference on Computer Vision and Pattern Recognition. 2021.

Other Concerns:
- Please refer to the supplementary material in the main text for the definition of LDDMM and spline deformation model or add a brief introduction to these models in order to be self-contained.
- How is the pipeline in Sec.4 (deep registration, D/RobOTP) trained? Did you train/pre-train each part individually or the full pipeline end-to-end? If so, how do you differentiate through the argmin in Alg.1? How stable is the training process? Is the blur radius and reach distance trained altogether?
- Line 98-100: I don't think the proposed projection is in -sharp contrast- with FLOT. FLOT utilizes a refinement module which learns the smoothness prior from the training data and it acts as an implicit regularization module compared to the explicit projection model proposed in this paper.
- According to Line194-Line203, the confidence weight reflects the occlusion status of the point clouds to be registered. Is this claim justified by the experiments? Please plot the confidence weight, e.g. in a representative sample in Flying3D dataset and verify the claim.
- The paper should note (and clarify) that the rigid projection in Eq.(5) is also related to a weighted Kabsch algorithm.
- Line 119: sorting to -> sorting in;  Fig.8 in Supplementary: fow -> flow;

**Time Spent Reviewing:**

5

---

> ### Author Response · Authors · 2021-08-10
> **Answer to Reviewer 2 - Part 1**
>
> We thank the reviewer for the time, advice and positive feedback. We will fix the typos and add the suggested references in the final version of our paper. We address and summarize the main questions below.
>
> ### How and why is the proposed method (weighted RobOT matching) faster and more robust than using the transport plan?
>
>
> Our key remark in Eq. (3) is that the optimal transport plan $\pi_{i,j}$ has a simple "symbolic" structure that is a perfect fit for the KeOps library. In practice, on top of the reduced memory usage (from $O(N^2)$ to $O(N)$), **we observe a sizeable speed-up (x10-x100) when compared with a cuBLAS baseline for the same computation.**
> We refer to "Fast geometric learning with symbolic matrices", Feydy et al. (NeurIPS 2020) for details on the low-level CUDA schemes (with e.g. optimizations of register usage) that make these improvements possible.
>
> In practice, relying on KeOps "symbolic tensors" to turn the output of the GeomLoss solvers into a weighted vector field is thus more efficient than using either a full transport matrix or a sparse tensor with e.g. the top-10 values per row of the transport plan.
>
>
> In terms of **robustness**, there are three main advantages to our "weighted RobOT matching" of Eq. (4):
>
> - It is fully compatible with the **modern theory of unbalanced OT** -- which takes outliers and occlusions into account.
> - We can implement it using the fast Log-Sum-Exp reductions that are provided by the KeOps library, thus **avoiding the numerical overflows/underflows** that limit usual implementations of the Sinkhorn method to large values of the regularization parameter $\varepsilon$.
> - Unlike previous formulations that relied on the gradient of the OT loss to drive the matching (including e.g. the work of Jean Feydy, the author of the GeomLoss solvers), our vector field $v$ always has a **homogeneous magnitude**. Whereas gradient-based methods under-estimate the OT displacement $v_i$ of points $x_i$ such that $\sum_j \pi_{i,j} < \alpha_i$ and over-estimate it when $\sum_j \pi_{i,j} > \alpha_i$, our method takes this effect into account with an appropriate normalization and importance weight - Eq. (4). We note that as detailed in Lines 151-153 of our paper and illustrated in reference [83], this "inhomogeneous gradient" problem was a well-known issue of unbalanced OT methods for shape analysis.
>
>
> We will make these points clearer in the final version of the paper. Overall, our formulation is scalable, versatile and **easy to interface** with existing registration models as a **plug-in replacement** for nearest neighbor projection.
>
>
>
>
> ### The size of the test set (10 lung pairs) is too small when compared to the size of the training dataset (1,000 lung pairs).
>
> The 10 lung pairs used for testing come from the COPDGene DirLab dataset -- a standard benchmark in the field. For this dataset, experts annotated 300 corresponding landmarks per image-pair. Since annotating this data is a costly and labor-intensive task, only these 10 pairs are currently available to us for testing. (As an example of another recent annotated dataset for lung registration, see e.g. the 2021 Learn2Reg challenge that provides no more than 30 lung pairs -- https://learn2reg.grand-challenge.org/.)
>
> Our contribution has been to extend this (small) standard dataset with an **additional 1,000 point cloud pairs for which we do not have this pointwise correspondence information**, but which we can use for training. As detailed in Lines 219-229 and in Section A.3, the fact that we do not have correspondences for these cases means that our approaches are trained entirely on *simulated* deformations: we do not use or require expert-provided corresponding landmarks for our training.
>
> In this context, our testing evaluation on the 10 COPDGene DirLab pairs should be understood as evidence for the ability of our model to **generalize from synthetic deformations to genuine breathing movements**. These 10 pairs and their costly annotations were not used for training or validation, but strictly for testing.
>
> ### What about "Weakly Supervised Learning of Rigid 3D Scene Flow" and "HCRF-Flow: Scene Flow from Point Clouds with Continuous High-order CRFs and Position-aware Flow Embedding" (both published at CVPR 2021)?
>
> We thank the reviewer for letting us know about those two relevant and very recent papers (we note that they were presented at CVPR 2021, three weeks after the deadline for NeurIPS submissions).
>
> We will include both papers in our discussion and in our Kitti benchmarks, Figure 4. We summarize our main comments below:
>
> 1. **"Weakly supervised..."** focuses on rigid and rigid-by-part motion estimation. This method is tailored for autonomous driving, with a strong inductive prior. It relies on a segmentation of the input point cloud between background (road...) and foreground (cars...) regions:
>   - On the one hand, the authors apply the vanilla Sinkhorn algorithm (with an extra "dustbin" for outliers) to background points to estimate the global (rigid) camera motion between two time steps. This essentially corresponds to the "Robust Point Matching" algorithm (Gold et al. 1998, reference [41] in our paper), a classic method in the field.
>   - On the other hand, the movement of individual objects in the foreground (e.g. cars) is explained by independent rigid motions.
>
> 2. **"HCRF-Flow ..."** proposes a "post-processing" refinement step. The pipeline has similarities with our D-RobOT model, even though the theoretical derivations are entirely different:
>   - The HCRF model relies on the solution of a non-convex optimization problem that promotes local rigidity on supervoxels, which is computed using an iterative message passing algorithm.
>   - Our RobOT finetuning relies on the solution of an optimal transport problem that is followed by e.g. a spline regularization -- two standard operations for which scalable solvers now exist.
>
>
>
> In comparison with our work, we first note that **our method significantly outperforms both approaches on the Kitti dataset**. After training on the Flying3D dataset, testing on the 142 Kitti scene pairs sampled with 8,192 points each, **"Weakly ..."** reports an EPE3D of **42mm** (lower is better) and an Acc3DS of **84.9%** (higher is better). **"HCRF-Flow"** reports an EPE3D of **53mm** and an Acc3DS of **86.3%**.
> Meanwhile, in the exact same setting, our **D-RobOT** method with **spline** regularization reaches an EPE3D of **33mm** and an Acc3DS of **90.4%**. Since our method is scalable, we also report results with 30k points per shape ("dense" labels in our tables), reaching an EPE3D of **22mm** and an Acc3DS of **95.9%**.
>
>
> Second, and more importantly, we believe that **our paper has a wider appeal than both of these methods**.
>
> Instead of focusing on e.g. scene flow estimation, we attempt to bring together the applied maths and computer vision communities by **taking a higher-level perspective** on 3D shape registration. As detailed in our answer to Reviewer 3 (Y4k6), our paper shows that Robust OT is a versatile and scalable tool. It brings significant improvements to a very wide range of registration methods, from optimization-based models to deep neural networks.
>
> Notably, our RobOT approach **outperforms dedicated methods on Kitti** while also being able to **handle fully non-rigid problems** (such as breathing movement between lung vessel tress), being packaged as a **plug-in replacement for nearest neighbor projection** and resting on **solid mathematical foundations**.
> We thus believe that it will be of interest to the wider community and raise interest for modern OT theory in the computer vision literature.
>
>
>
>
>
> ### The authors should add a section on limitations, the broader impact statement and references to the definition of the LDDMM and spline models in the main body of the paper.
>
> We agree. Thanks to the extra page that is allowed for the final version of the paper, we will add these references in the main paper. Our paragraph on limitations and our broader impact statement will also be moved back to the main body of the paper.

---

> > ### Author Response · Authors · 2021-08-10
> > **Answer to Reviewer 2 - Part 2**
> >
> >
> > ### How is the deep registration pipeline (D-RobOT) pipeline trained? How do the authors differentiate through the argmin in Alg.1? How stable is the training process?
> >
> >
> > Algorithm 1 (optimization for RobOT-P) is not used by D-RobOT. To avoid confusion, let us stress the difference between our RobOT-P and D-RobOT methods:
> >
> > - **RobOT-P is an optimization-based approach** wich first deploys OT for feature matching and then regularizes these correspondences using a deformation model. The features are given a priori: they can be handcrafted (such as the FPFH descriptors) or learned (as shown in line 230).
> > - **D-RobOT is an end-to-end deep learning framework**, which takes a point cloud pair as input and outputs the deformed point cloud. It includes an OT based prealignment module (optimization based), a deep registration module (learning based, which **directly predicts** the parameters of e.g. a spline or LDDMM deformation layer) and a postprocessing module (optimization based).
> > The prealignment and postprocessing modules only consider xyz information and can be computed in milliseconds: we use them as plug-in modules in our pipeline. As a consequence, the training loss is only computed and backpropagated for the deep registration module: the training process is stable, without any backpropagation through an ArgMin layer.
> >
> >
> > We believe that both methods serve different purposes and will be of interest to different authors in the shape analysis community. In a nutshell: RobOT-P is easier to deploy with little to no training, while D-RobOT has a higher ceiling.
> >
> > Thanks to the reviewers' comments, we understand that more clarity is needed on our side: we will add an explicit discussion of the pros and cons of both approaches in our paper. As detailed in our answer to Reviewer 4 (Yt1x), we will also re-organize Sections 3 and 4 to make the paper easier to read.
> >
> > ### Are the blur radius and reach distance trained altogether?
> >
> > The blur and reach distances are constant hyper-parameters. As detailed in lines 129-141, their values can be chosen using simple heuristics: precise values for all of our experiments are given in the Supplementary Material.
> >
> > ### I don't think that the proposed projection is in *sharp contrast* with FLOT, which relies on an implicit regularization prior.
> >
> > We understand that opinions on this point may differ, depending on the target application. Unlike implicit methods, *explicit* regularization priors provide **strong mathematical guarantees** on the smoothness and invertibility of the final deformation -- which can be a necessary requirement for downstream applications.
> >
> > For instance, in the context of lung registration, the fact that our LDDMM-based methods (both RobOT-P and D-RobOT) are **guaranteed to produce diffeomorphic registrations for the breathing motion** is a key selling point. In a clinical scenario, displaying a non-diffeomorphic registration result to a pulmonologist would dramatically hamper the credibility of our model.
> >
> > We will make this point clearer in the paper and choose a more nuanced wording.
> >
> >
> > ### Is the claim that "the confidence weight reflects the occlusion status of the point clouds to be registered" justified by the experiments? Illustrations on the Flying3D dataset would be appreciated.
> >
> > Indeed, we observe this desirable behavior in our experiments. This makes sense, since salient features that are occluded in one shape but not in the other may not find a good "match" in a $\tau$-neighborhood in the feature space (where $\tau$ is the "reach" distance).
> >
> > We thank the reviewer for this remark and will add the suggested visualizations in the Supplementary Materials.

---

### Official Review · Reviewer_Cd96 · 2021-07-23

**Rating:** 7
**Confidence:** 3

**Summary:**

This paper proposes to combine optimal control (OT) and a set of deformation models for point cloud registration. They propose two settings to leverage OT: global feature matching and deep feature prediction. Experiments on two datasets verified the effectiveness of the proposed method. The writing is overall clear.

**Limitations And Societal Impact:**

Yes.

**Main Review:**

This paper proposes to combine optimal control and a set of deformation models (e.g., rigid, affine, spline, and LDDMM) for point cloud registration. The bijectivity from the optimal control can provide reliable matchings and gradients. The bijective matching is typically time-consuming, but the proposed method leverages an efficient OP solver and scales to more than 10k points.

The method proposes two settings to utilize OT for point cloud registration:
The first one is global feature matching. The feature may come from heuristic rules of neural networks. They then use the extracted feature to solve a "weighted robOT matching". They then project the result with certain deformation models.
The second one is registration via deep deformation prediction:
They first use a fast OT solver for rigid or affine pre-alignment. They then propose to use a network to predict the parameters of a chosen registration model. The model is trained with supervision. Finally, they RobOT to fine-tune and post-process the result.

The proposed methods achieve the SOTA results on two datasets: lung registration and the Kitti dataset.

The overall idea is novel, clear, and effective.


---------Weakness:

For partial registration, there is only one toy example shown in supplementary. It would be helpful to show more results on incomplete and occluded point clouds. For example, we can have large-scale quantitative experiments on the ShapeNet dataset. The task can be registering two random views, which is of great interest to many readers.

Deep-feature learning: Using cross-entropy to force the feature to preserve Euclidean distance may not be the best solution. Another idea would be contrastive learning with matched pairs as positive pairs and others as negative pairs.

For Kitti experiments, there aren't many qualitative results. It would be better to include more visualization for local matching details, as many readers may be interested in the Kitti experiments.

The author claims that "the image-based methods can make use of more information." It's interesting to know whether the proposed method will outperform the image-based methods when the color information is taken as input.

Symbols of the deep-feature learning part (starts from line 230) are not very clear. More explanations are encouraged.

In figure 3, for the optimization-based method, it's not clear whether the runtime includes the deep-feature inference time.

In Kitti experiments, it's not clear which feature is used for matching. Handcrafted feature of NN feature?

The black-box optimization is still time-consuming. For example, LDDMM takes 40 seconds.

MSN[1] also provides an EMD implementation that supports more than 10k points without memory issues. Please discuss the work.

[1] Liu, Minghua, et al. "Morphing and sampling network for dense point cloud completion." Proceedings of the AAAI conference on artificial intelligence. Vol. 34. No. 07. 2020.

**Time Spent Reviewing:**

4

---

> ### Author Response · Authors · 2021-08-10
> **Answer to Reviewer 1 - Part 1**
>
> We thank the reviewer for the time, advice and positive feedback. We address the main questions below:
>
> ### Could the authors show more examples for partial registration, e.g. between multiple views on the ShapeNet dataset?
>
> This is in an excellent suggestion. In the supplementary material, we will include illustrative figures and a benchmark table for this standard problem. To keep things simple, we will build upon our example for the Stanford Bunny and restrict this study to the standard FPFH features that we discussed in Suppl. A.1.
>
> Going further, studying learned features and deep prediction for multi-view reconstruction will be the subject of future work: we also believe that designing registration methods that are robust to partial acquisitions is an important research direction.
>
>
> ### Why not use a contrastive loss for the deep feature learning?
>
> Using a contrastive loss is a good suggestion.
> We note a small caveat however, which is that contrastive losses require positive and negative samples. For point cloud registration, a strict distinction between "positive" and "negative" samples may be too strong since we may still want to promote the **registration of a point to a $\sigma$-neighbor of its corresponding point**.
>
>
> Fortunately, in our case, the lung registration task is a fairly "closed" and well-defined question -- compared with metric learning in open world. This allows us to use a strong loss in our approach, as **we compare the correct mapping to all the others.  This is a one-vs-rest strategy**.
>
> Instead of sampling positive and negative samples (as in a contrastive loss), for a synthetic pair of point clouds with $N$ points in correspondence with each other, we construct $N\times N$ **correspondence matrices that take all possible point pairs into account** ($N=60,000$ in our experiments). The KeOps library allows us to manipulate these objects efficiently to compute the required losses and gradients, with extremely fast run times and without memory overflows.
>
> Specifically, our loss for feature learning works as follows:
>
> 1. Assume that we are given two point clouds $x_1, \dots, x_N$ and $y_1, \dots, y_N$ that are in pairwise correspondence with each other. In our experiments, these are typically the output of a synthetic "data augmentation" procedure as for the Flying3D objects (for the Kitti benchmark) and our synthetic lung pairs (for the Dirlab benchmark).
> 2. Apply the feature extractor (a trainable neural network) on both point clouds, independently from each other. We retrieve point features $p_i$ and $q_i$ that are respectively associated to the $x_i$'s and $y_i$'s.
> 3. Compute the $N\times N$ correspondence  matrix for the point positions in the **source point cloud** $c_{(x_i,x_j)} = \text{softmax} (\beta\|x_i-x_j\|^2_{\mathbb{R}^3})$ $ = \frac{\exp(-\beta\|x_i-x_j\|^2_{\mathbb{R}^3})}{\sum_{j=1}^N\exp(-\beta\|x_i-x_j\|^2_{\mathbb{R}^3})}$  where the Softmax denotes a mirrored exponential followed by a normalization over the lines of the correspondence matrix while $\beta$ is a scaling factor that determines the ambiguity that we can tolerate. Each row of this matrix then refers to a probability distribution, a position heatmap with a peak at point $x_i$. The peakiness is determined by the hyper-parameter $\beta > 0$.
> 4. Similarly, we compute a correspondence matrix for the **source and target features** $c_{(p_i,q_j)} = \text{softmax}(\|p_i-q_j\|^2_{\mathbb{R}^\text{D}})$, where as above each row refers to a feature heatmap now indicating how well $p_i$ corresponds to $q_j$.
> 5. Our total loss is the sum over the cross entropies for each row: $\text{CE}(c_{(x_i,x_j)}, c_{(p_i,q_j)})=-\sum_{i=1}^N \sum_{j=1}^N c_{(x_i,x_j)}\log~c_{(p_i,q_j)}$.
>
>
> For each pair of points $(x_i, y_j)$ that may or may not be in correspondence with each other, our softmax loss thus promotes the correct correspondence ("$q_j$ should be close to $p_i$ if $x_j$ is $\beta^{-1/2}$-close to $x_i$ i.e. if $y_j$ nearly corresponds to $x_i$") while discouraging *all the other* correspondences in a smooth and differentiable way.
>
>
> ### Could the authors explain their notations for feature learning around line 230?
>
> We thank the reviewer for pointing out the ambiguity in our notations -- this will be clarified in the final version of our paper, following the discussion above.
>
>
>
> ### Could the authors add qualitative visualizations for the Kitti dataset?
>
> We agree that such an illustration will be a welcome addition to our paper. Thanks to the extra page that is allowed at the end of the review process, we will add a Figure in the main paper to supplement our quantitative results in Figure 4. We will also include more detailed visualizations in the supplementary material.
>
> ### It will be interesting to see whether the proposed method will outperform the image-based methods when the color information is taken as input.
>
> Indeed. Thank you for your interest in our method: we are actively working on this topic.
>
> So far, we have shown that "purely geometric" descriptors can already result in an accuracy that is on the order of a millimeter for the complex lung registration problem -- a performance which is of interest to the wider optimal transport, geometric deep learning and 3D vision communities.
>
> Going forward, our next step (and main clinical target) is to reach "pixel-perfect" accuracy by adding image-based information to our 3D points.
>
> ### In Figure 3, do the run times for the optimization-based methods include the deep-feature inference time?
>
> Yes. The runtime of the entire pipeline is included in the run times. We will make this clearer in the paper.
>
>
> ### The black-box optimization is still time-consuming: for example, LDDMM takes 40 seconds.
>
> We fully agree. We included these costly optimization-based results as a reference for authors in computational anatomy. In this field, optimization-based LDDMM pipelines still provide a widely accepted state-of-the-art baseline in terms of accuracy.
>
> Going further, we believe that these slow run times are limiting for large-scale applications. Improving on these run times without compromising on accuracy and guarantees for the deformations is our main motivation for the development of a deep *prediction* framework, where **this costly numerical optimization is replaced by prediction** at test time.
>
> As detailed in Figure 3, our best performing method for the lung registration task - DRobOT (LDDMM) - runs in less than 2s on high-resolution point clouds and outperforms its optimization-based counterpart - RobOTP (LDDMM) - by a significant margin.
> For 3D shape registration in medical imaging, the way forward is clearly to replace black-box optimization modules by deep predictors for the parameters of the deformation model. (In the case of LDDMM: a vector field, the so-called "shooting momentum".)
>
> ### For matching on the Kitti dataset, did the authors use handcrafted or neural features?
>
> We perform two main experiments on the Kitti dataset, **both of which only use the xyz point positions as input**:
>
> 1. The optimization-based RobOT-P method takes the xyz positions  as input and solves an optimal transport problem. The results can be found in the left table in Fig.4 (the last two rows in "Unsupervised" section). Interestingly, this strategy already outperforms most supervised deep learning approaches in terms of accuracy, speed and memory consumption.
>
> 2. The other experiment uses the (deep) prediction-based D-RobOT method. Compared with the explicit feature matching used in RobOT-P, this second approach relies on an implicit matching of features that we describe in Sec. 4.1. The full D-RobOT model includes (a) a fast rigid prealignment (optimization based), (b) a deep registration module (learning based), and \(c\) a fast postprocessing (optimization based). The three successive steps (a), (b) and \(c\) only take as input the xyz point positions of the moving and target point clouds.

---

> > ### Author Response · Authors · 2021-08-10
> > **Answer to Reviewer 1 - Part 2**
> >
> >
> > ### Could the authors discuss the EMD (= OT) implementation of "Morphing and sampling network for dense point cloud completion", Liu et al. 2020?
> >
> > We thank the reviewer for this interesting reference that we will add in our literature review. In this paper, the authors rely on a GPU implementation of the classic **Auction algorithm** to solve the OT problem approximately with O(N) memory footprint. **This is a good method, but it has the following drawbacks:**
> >
> > - For small values of the regularization parameter epsilon, which are required to produce accurate registrations in 3D, the algorithm may take **thousands of iterations** to converge. In practice, the reference implementation for this paper (https://github.com/Colin97/MSN-Point-Cloud-Completion) uses 10,000 iterations for testing (with eps = 2e-3) which is prohibitive for our use case. At training time, one is thus required to (massively) compromise on accuracy to obtain acceptable run times: in this work, the authors terminate the auction loop after 50 iterations and use a larger value for the regularization parameter (eps = 5e-3). This compromise may be acceptable when OT is merely used to compute an EMD loss and gradient during the training of a neural network. In our case however, this is not acceptable: the accuracy of the OT solution has a direct impact on the quality of the final matching. We thus need a solver that is both **fast** (to keep total run times on the order of a second) and **accurate** (to reach a precision on the order of a millimeter, without over-smoothing).
> > - The proposed GPU implementation of the auction algorithm is **limited to the "bijective" case**: balanced OT with the same number of points in both shapes and uniform weights per points.
> >
> > To go beyond these limitations, our work relies on the much more recent **multiscale Sinkhorn** solvers that are provided by the GeomLoss library. This allows us to:
> >
> > - Solve OT problems with **high accuracy** using no more than **10-20 iterations** of the (multiscale) Sinkhorn algorithm. We note that the complexity of these iterations is comparable to those of the auction algorithm (the main difference is that the "minimum" in the auction iterations is replaced by a "minus log-sum-exp", i.e. a "softmin" operator).
> > - Handle shapes that are sampled with an **arbitrary number of points**, possibly with importance weights (as in the lung registration case).
> > - Rely on the general theory of **"unbalanced" optimal transport**, which has been developed since 2016 and allows us to be more robust to outliers and partial views.
> >
> >
> >
> > For an in-depth discussion of the x100-x1,000 speed-up that is provided by the multiscale Sinkhorn solvers when compared with the classic Auction and Sinkhorn algorithms, we refer to Section 3.3 of Jean Feydy's PhD thesis (http://jeanfeydy.com/geometric_data_analysis.pdf) - this document summarizes the mathematical documentation of the GeomLoss library. Algorithms 3.2-3.3 (Auction + Sinkhorn algorithms) and Figure 3.29 (extensive benchmarks) may be especially relevant to the reviewer's comment.

---

### Decision · Program_Chairs · 2021-09-27

**Decision:**

Accept (Poster)

**Comment:**

This paper was a borderline case, but despite some weaknesses we are recommending acceptance for NeurIPS.  Here, the positive aspects are mostly practical/empirical; the authors show how unbalanced OT can be incorporated into realistic pipelines for rigid/nonrigid registration.  But the work is mostly engineering, and the methodological/algorithmic contribution is smaller.

The AC/SAC also had some concern about precisely what objective function this algorithm is optimizing.  It is not articulated clearly in the paper and appears to be a bilevel problem with optimal transport as the "inner" problem----but then the gradients may not be correct.  Empirically this appears to be OK in the results, but it'd be preferable to give a clearer mathematical story in this work.

Two additional related papers need to be acknowledged/discussed in the final version of this paper:
* Feydy et al.  "Optimal Transport for Diffeomorphic Registration."  MICCAI 2017. ---- This paper contains many related ideas and incorporates optimal transport losses in a similar fashion.
* Mukherjee et al.  "Outlier-Robust Optimal Transport."  ICML 2021. --- This paper proposes a similar unbalanced transport model.  It also uses the same "ROBOT" acronym to refer to their model.
Please make sure the camera-ready acknowledges and discusses these two works, and also take the suggestions in the individual reviews seriously.